# Passive Dynamics of the Head, Neck and Forelimb in Equine Foetuses—An Observational Study

**DOI:** 10.3390/ani13121894

**Published:** 2023-06-06

**Authors:** Carla M. Lusi, Helen M. S. Davies

**Affiliations:** Department of Veterinary BioSciences, The University of Melbourne, Parkville, VIC 3010, Australia; clusi28@gmail.com

**Keywords:** passive dynamics, equine, forelimb, head, neck, lameness

## Abstract

**Simple Summary:**

In horses, the whole body and the individual limbs move in particular patterns that are constrained by the connections between the different parts. In this study of dead foals, we demonstrate that the movements of the front limbs forwards and backwards cause specific and consistent movements in the head and neck. In contrast, similar movements in the head and neck cause no such movements in the limbs. This unidirectional mechanical connection is different from the bidirectional reciprocal movements within the forelimb that constrain the limb to work as a whole. Such mechanical effects of limb movement on the head and neck may integrate with the nervous controls to provide near-instantaneous adjustments to postural disturbances, thus promoting rapid and efficient locomotion.

**Abstract:**

Passive dynamics is an aspect of locomotion which is entirely dependent on the mechanical configuration and linkages of adjacent body segments. Tension distribution along mechanical linkages enables the execution of movement patterns with reduced need for complex neurological pathways and may play a role in reestablishing postural stability following external disturbances. Here we demonstrate a uni-directional mechanical relationship between the equine forelimb, head and neck, which may have implications for balance and forelimb loading in the horse. These observations suggest that forelimb, head and neck movement coordination (observed in the horse during unrestrained locomotion) is significantly influenced by the mechanical linkages between body segments, rather than being entirely dependent on neurological input as previously thought. This highlights the potential significance of research directed at investigating passively induced movements in understanding common locomotory patterns. Additionally, it suggests a mode of postural control which may provide instantaneous adjustments to postural disturbances, thus promoting rapid and efficient locomotion.

## 1. Introduction

In the existing literature, two main modes of postural stability and motor control are established, including reflexive neural feedback and anticipatory systemic changes of posture [1,2,3]. However, a third mode which places emphasis on the mechanical reactions and passive dynamics of linked segments in the body can be found throughout the literature [3,4,5]. It has been suggested that complex and skilled movements are simply and effectively controlled via the utilization of passive forces [6] and that the distribution of these forces (through anatomical or mechanical coupling) further simplifies postural and movement control by reducing the degrees of freedom the nervous system has to consider [6,7,8].

In horses, research into the role of passive dynamics in maintaining balance and locomotory patterns has mostly been approached from one of two directions. The first of these considers the passive interaction of joint segments through the storage and release of elastic energy [9,10,11,12,13,14]. This has been most clearly demonstrated in the biomechanics of the forelimb [14,15,16]. The second approach relates to the role of visual, vestibular and somatosensory inputs in enabling segmental movement and contributing to whole body balance and spatial orientation [17,18,19,20,21]. Ultimately, it has been demonstrated that there are mechanisms of motor control, which rely on a sophisticated interaction between labyrinth and neck reflexes and serve to stabilize positions of the head, neck, trunk and limbs in order to compensate for asymmetrical loads or forces.

Although both research directions have been useful in understanding forelimb and whole-body biomechanics in horses, it is noted here that a functional link exists between the two research areas that is yet to be explored. The investigation of energy storage and release through biological spring systems is essentially limited to the mechanical connections of the musculotendinous unit. Hence, passive movements enabled through elastic energy storage are confined to the segments across which the musculotendinous unit spans. On the other hand, segmental mechanics through nervous input do focus on the movement and coordination across several body segments (such as the head, neck and trunk), yet it does not investigate the mechanical link between these segments.

Hence, it is the purpose of this present study to investigate the mechanical interaction between body segments in the horse which may contribute to the efficiency of locomotion. The relationship between head and forelimb movement presents as a prominent non-volitional movement pattern where passive forces between mechanically linked segments may dominate and exist independently of neurological input. Studies looking at head and limb movement synchronization in other species suggest that neural reflex mechanisms play a large part [22,23] in coordination of body segments; however, several other more recent studies suggest that intersegmental dynamics may dominate in this role [4,5,24,25,26]. If this is the case, mechanical linkages would need to be independent of feeding and other neural based responses instituted by the main exteroceptors like the eyes and ears in order to avoid interference with body coordination and locomotion from such movements of the head and neck. Therefore, it is hypothesized here that a uni-directional mechanical relationship exists between the head and limb. It is further hypothesized that the directions of related movements are consistent. This sort of mechanical predictability may provide a simplified mechanism of movement coordination and therefore makes redundant the need for complex neurological reflex pathways from the distal forelimbs for such coordination.

## 2. Materials and Methods

Sixteen foetal foal cadavers (estimated ages 80–340 days post coitum [27]) of various breeds were collected from pregnant mares that had been euthanized for reasons not associated with this study. A foal that had died within 24 h post-partum was also collected and investigated as part of the study. All foals appeared to be of normal conformation and stage of development for their gestational age with no abnormalities (such as flexural deformities) observed.

Foal cadavers that were not investigated immediately after collection were stored at −20 °C until needed. In preparation for dissection, they were thawed at −4 °C for 4–7 days depending on their size. Once thawed, foals were placed in lateral recumbency on a smooth, metal surface with their limbs positioned to approximate a standing position and their head placed in a natural position as would be observed when the head and neck are unrestrained. The surface under the head and neck of the horse was wet sufficiently with water to reduce friction and to allow freedom of movement prior to the commencement of video recording. A Sony NEX-5 digital camera was positioned on a tripod at a distance of approximately 1 m above the foal to photograph and record observed movements between the forelimb, head and neck. This camera remained in the same position for the entire investigation.

The primary investigator protracted and retracted the uppermost forelimb within its normal physiological range. Maximum protraction and retraction were taken as the most cranial and caudal points, respectively, to which the forelimb could be moved without influencing the position of the thorax and hindlimbs. This was conducted first with a natural degree of flexion allowed at the carpus (i.e., no force applied to the carpus) and then with the carpus maintained in maximum extension (obtained by placing pressure on the dorsal aspect of the carpus). Following this, the limbs were returned to the natural position described above, and the head and neck were moved repeatedly between a flexed and extended position (of various degrees) to observe any related movements produced in either the limbs or the trunk. The direction of each movement was noted and the presence or absence of effects distributing to other body segments with each movement of the head and neck or limb. The effect of these movements in both left and right lateral recumbency were tested in the specimens, and independent observers gave their opinion on the presence or absence of related movements and the direction of any movements.

Eleven of the seventeen subjects investigated were then skinned, and the same sequence of forelimb and head and neck movements were carried out. In three foetuses, the left forelimb and the cervical musculature on the left side were completely removed to reveal the skeleton and to observe any thoracic and spinal movement accompanying protraction and retraction of the remaining right forelimb (Figure 1).

In three subjects where the upper limb was removed, there was sequential removal of most of the extrinsic shoulder muscles (latissimus dorsi, brachiocephalicus and omotransversarius, the pectoral group and trapezius and rhomboideus) in various sequences and sequential removal of the forelimb from distal to proximal until just the dorsal half of the scapula and the serratus ventralis muscle remained. During these sequences, either the forelimb or its remnants were protracted and retracted, and any thoracic or spinal movements related to those movements were recorded.

## 3. Results

Despite differences in ages, breeds and genders, a fundamental and consistent uni-directional mechanical relationship was observed between forelimb and head and neck movement in all the specimens, and the movements were the same bilaterally.

Movement of the head and neck between a flexed and extended position did not produce any observable movement of the trunk or limbs. However, retraction of the forelimb caused simultaneous extension of the head and neck in all foals, which was most noticeable with pressure applied to the dorsal aspect of the carpus (to maintain straightness of the limb). In subjects that were skinned and/or had the left forelimb and cervical musculature removed, forelimb retraction was also observed to cause substantial movement of the thoracic cage in the dorsocaudal direction (Figure 2). This, in turn, produced dorsiflexion (extension) of the thoracic and cervical spine.

In contrast, protraction of the forelimb resulted in simultaneous head and neck flexion in all foals. Effects on the position of the thoracic cage were not as clearly evident as they were with forelimb retraction; however, cervical-thoracic ventriflexion (spinal flexion) was clearly observed in specimens where cervical musculature had been removed (Appendix A). The greatest range of axial skeletal movement related to protraction or retraction of the forelimb appeared to be related to the caudal cervical vertebrae and the cranial thorax.

Passive movements and tension distribution caused by flexion and extension movements in individual forelimb joints in foetal foals demonstrated consistent bidirectional patterns and relationships within the limb. One of the most obvious of these relationships occurred between the carpal, fetlock, elbow and shoulder joints. Extension of the fetlock joint locked the carpus in extension and directed tension so that the elbow and scapulohumeral joints extended simultaneously. Demonstrating the bidirectional relationship, flexion of the carpus could then not occur without simultaneous flexion of the elbow and scapulohumeral joints. Similarly, extension of the carpus whilst the limb was protracted was not possible with the elbow joint flexed, meaning that the elbow joint could not be flexed without simultaneous flexion of the carpus. Overall, such connections and relationships ensure that the limb functions mechanically as a single entity despite the degree of disconnection caused by its comprising joints. Generally, it was observed that, in comparison to the tensional lines observed in a ventrally extended position, those observed in a protracted position were more pronounced, whilst those observed in a slightly retracted position were not obviously changed.

There were no observable differences in any of these movement patterns when placing foetuses in either left or right lateral recumbency nor were there any grossly observable differences noted after removal of skin (in foetuses where this was applicable).

Sequential dissection of the extrinsic muscles of the shoulder and removal of the limb from distal to proximal demonstrated that the unidirectional relationship between the forelimb, neck and head persisted until the dorsal part of the scapula and its attachment to the serratus ventralis was removed. Until that point, movement of the scapula within its normal craniocaudal range created consistent flexion and extension of the head and neck. In contrast, movements of the head and neck did not affect the remaining parts of the limb as they were absorbed into the serratus ventralis muscle. Movements of the remaining forelimb consistently created movements of the head and neck as described above.

## 4. Discussion

This study set out to test the hypotheses that a uni-directional mechanical relationship exists between the equine head and forelimb and that this relationship is consistent. An inference from this would be that the coordinated movement observed between the head, neck and forelimb in the horse during locomotion is likely to be somewhat dependent on the mechanical connectivity between segments rather than being entirely dependent on neurological input. Results from this study confirmed the basic hypotheses by identifying synchronous uni-directional consistent locomotory patterns between the forelimb, head and neck in equine foetal foal cadavers. To our knowledge, this is the first example of unidirectional anatomically related movements which cannot be directly explained by reflexive neural feedback mechanisms. Instead, it is concluded here that coordinated forelimb, head and neck movement in the equine foetus can be achieved through passive dynamics.

The use of foetal foal cadavers to investigate the tensional distribution of forces throughout the forelimb fascia and related movements throughout the foetus was decided on for a number of reasons. First and foremost, the size and relative weight of foetal foal cadavers made limb movements throughout the study much more manageable. Secondly, embryonic and developmental research in equids has demonstrated that collagenous fascial and muscular connections arise from a structural network that is developed very early on in utero [28,29,30]. This strongly suggests that the related movements through the fascial network in foetal foal cadavers is a valid representation of that in live adult horses. Admittedly, it is almost certain that changes will occur to the fascia with post-partum loading, as fascial architecture has been shown to adapt structurally to functional loads [31]. However, it is reasoned that investigation of the fascial anatomy prior to any loading-induced changes provides a basic anatomical foundation necessary for understanding the overall functional significance of fascia and its role in passive dynamics.

The large range of gestational age and condition of the foetuses as well as movement and drying of the water film underneath the specimens meant that absolute measurements of the size of movements were very variable. Repeated movements always created the same directional effects. Independent observers observed the same directional relationships and proportionate effects.

The concept of intersegmental passive dynamics has most clearly been demonstrated in the study of robotics [25,32,33,34,35]. Mechanical bipedal robots (lacking any active control system) have been shown to settle into a steady state of walking which is maintained solely by the passive forces acting between segments [25,33,34]. This implies a transferral of forces (ground reaction and gravitational) acting on the limb, which is dependent solely on the mechanical configuration of the robot.

Observations from this present study demonstrate a similar mechanically dependent passive interaction between the head, neck and forelimb of foetal foal cadavers, which correlates with synchronous head, neck and forelimb movement patterns observed in the live horse at a walk. As described above, application of an external force to the forelimb to replicate forelimb protraction (or retraction) produced simultaneous head and neck flexion (or extension) without any other external forces being applied to the head, neck or any other body part. This suggests that, in equine foetal foal cadavers, forces applied to the distal forelimb in the sagittal plane are distributed through the fascial and musculoskeletal elements of the forelimb to the shoulder girdle, neck and head.

The findings from this study provide support for previous studies on humans, which have presented hypotheses on the role of passive dynamics in human locomotion but have been unable to irrefutably attribute their observations exclusively to the mechanical interaction of segments [3,4,5,26]. By using foal cadavers, it has been possible in this study to identify the relative degree of synchronous head, neck and forelimb movement, which is executed independently of neural input in the equine foetus or neonate. Additionally, it has enabled a clear demonstration of a passively generated uni-directional movement pattern that is functionally significant in the live adult horse and suggests that the study of passively related movements, which occur independently of any neural input, may assist in understanding equine biomechanics.

With this said, the importance of nervous system control is not undervalued. Research has demonstrated an interaction between labyrinth and neck reflexes which influences limb stance and muscle tone in order to stabilize the position of the trunk and maintain the animal’s centre of mass (COM) [20]. In horses, the mechanical relationship between the head, neck and forelimbs may further contribute to the maintenance or repositioning of their COM. Dagg (1962) [36] suggested that forward and backward head movement in horses throughout locomotion adjusts the horizontal COM position and mediates forward movement as a result. Our findings demonstrated that movement of the forelimbs can instigate this forward and backward head movement, suggesting that the passive relationship between the forelimb and head mediates weight transfer and perhaps minimizes the muscular effort required for propulsive forward movement. In response to this, it is proposed that labyrinth and neck reflexes, originating from the central control system in the head, instigate postural changes, which serve to re-stabilize the COM with movement. Hence, it appears that a complex and dynamic coupling exists between the nervous and passive dynamic systems of the body.

This then brings us to the next question raised from the observations of this present study, which relates to the directional control of each of these systems. Consistent external forces applied to the forelimb caused consistent coordinated movements of the head and neck, yet consistent external forces applied to the head and neck produced no observable effects on the limbs. Thus, it is hypothesized here that the nervous system implements control through connections, which develop outwardly from the head, whereas the mechanical system develops functional connections to the head and neck, which originate from the musculoskeletal elements of the limb and trunk. It is understood that the peripheral nervous pathways employed with reflexive postural adjustments are direct extensions of the neural crest derived structures in the head and neck, thereby supporting the first part of this hypothesis. However, more detailed studies looking at the development of the head, neck and limbs in various quadrupedal species is necessary to confirm the latter part of the hypothesis.

In contrast, movements within the forelimb are bidirectional where extension of any of the intrinsic joints (carpal, elbow and shoulder, digital or fetlock joints) create extension in all the others. In both ventrally extended and protracted positions, extension of the distal joints locked the carpal, elbow and shoulder joints in an extended position, thereby providing a passively coordinated structural support column. This is generally referred to as the passive stay apparatus [37,38,39]. When standing square in a live horse, passive support of the forelimb is enabled through ground reaction forces acting on the limb and causing fetlock hyperextension. The accessory check ligaments then act as tension bands for stability of the carpus, fetlock and digit, whilst several other musculotendinous structures passively extend the shoulder and elbow joints [38]. With forelimb protraction however, there is a lack of ground reaction forces which hyperextend the fetlock joint and maintain the carpus in a fully extended position. Instead, it can be assumed that active muscle contractility and/or passive forces from the catapult action of the limb [14] maintain its position. The locking of the carpus, elbow and shoulder joints in a protracted position would therefore contribute to the locomotory efficiency of the limb by providing inherent stability in preparation for impact.

This relationship between the intrinsic joints of the equine forelimb cannot be easily explained by the physical restriction of the articulating bones nor can it be explained solely by tension in the accessory check ligaments of the flexor tendons as described above. Desmotomy of the deep digital flexor tendon accessory ligament has been shown to cause redistribution of load in the forelimb without severely affecting the stability and overall locomotory ability of the limb [40]. This strongly suggests that other myofascial relationships are important in enabling the passive stay mechanism of the limb. In this study, it was observed that locking the carpus in extension (which occurs naturally through fetlock hyperextension) creates tension in the musculotendinous units spanning the flexor aspect of the limb, as well as the cranial and craniolateral aspects. Tension over the cranial and craniolateral aspects is particularly important because it explains how shoulder flexion is prevented. Lusi and Davies (2018) [41] showed how fascia covering the extensor carpi radialis attached proximally to the lateral humeral shaft. In addition to this, we know from early anatomical works that the extensor carpi radialis provides insertion for the distal biceps brachii tendon (the lacertus fibrosis) [39,42] which in turn transmits tension through the biceps brachii. The origin of the biceps brachii on the tuber of the scapula [39] allows passive resistance to shoulder flexion through tension in the biceps brachii. It is suggested here that it is also through the extensor carpi radialis that elbow joint flexion is largely restricted when the carpus is extended. Furthermore, a particularly intricate arrangement of fascia was observed in the dissections of adult horse limbs which neatly and strongly connected muscles of the shoulder, elbow and antebrachium [41]. Such connectivity strongly suggests that the architectural arrangement of the fascia is designed to coordinate these limb segments thereby contributing to the passive support of the limb.

Sequential dissection demonstrated that it was the movements of serratus ventralis muscle and its connections between the scapula, the caudal cervical vertebrae and the cranial ribs that was primarily responsible for transmitting the forelimb protraction and retraction movements into the head and neck. This suggests that interference to the normal movement of that muscle or its connections may have consequences for the overall control and speed of reaction to disturbances of balance in the forelimbs. Such disturbances of control might be expected to significantly change the forces on the caudal neck vertebrae where the serratus ventralis muscle attaches and where the greatest range of movements was observed with the forelimb movements. Hence, the persistence of such a disturbed movement might be expected to have consequences for the bone shape and bone quality in that region of the neck. As foetal and neonate cadavers were used, this study could not confirm that these movements were exactly what would be expected in adult living horses, nor that such a neurological boost to the speed of response to changes in limb movements exists. However, the functional anatomical connections are highly suggestive of the importance of this relationship between the forelimb and caudal neck movements in the maintenance of soundness in this region of the skeleton. Finally, it is suggested here that findings and hypotheses formed from this study on foetal foal cadavers can be extrapolated to live adult horses. As alluded to above, research on the development of connective tissue in utero suggests that the anatomical relationships observed in the foetuses of this study are indeed applicable to adult horses, as the collagenous fascial and muscular connections responsible for connected movements arise from a structural network developed very early on in utero [28,29,30]. Additionally, in cases where the foals underwent a freeze–thaw cycle, it was assumed that the passive movements observed in this study were not significantly altered by this. Past research investigating the effects of multiple freeze-thaw cycles on joint movement and tensile behaviour in soft connective tissue structures have demonstrated single freeze–thaw cycles to have no significant effect on the movement or biomechanical properties of these structures [43,44,45]. Further, the connected movements occurred in all foetuses and were completely consistent between foetuses of different stages of development and breeds. Hence, the mechanical relationship between the head, neck and forelimb that are obvious in foetal foal cadavers is suggested here to be a valid representation of a passive dynamic system in the live adult horse, which likely has significant implications for the efficiency and control of movement.

In post-mortem observations of very fresh intact adult equine cadavers (no rigor mortis) that are small enough to manipulate, movements of the forelimb create the same directional effects on the head as those observed in the foetuses (personal observation by HD). Similarly, in living horses the movements of the caudal neck in relation to the limbs can be clearly observed in unconstrained horses moving at speed over uneven terrain as in the film “The Silver Brumby” [46].

Admittedly, the effects of gravity and load bearing on the forelimb and their effects on the observed connectivity between the forelimb, head and neck were not able to be measured with this study design. However, given that there have already been studies investigating some degree of relationship between forelimb, head and neck movements in live, weight-bearing adult horses [19,47,48], we propose that the mechanically derived forward and backward motion of the head demonstrated in our study will persist when these forces are in play.

The clear movements of the caudal neck and cranial thoracic region in unconstrained horses moving freely suggest that such connected movements are important in the maintenance of normal use and loading of the bones, joints and related tissues in that region. If this is so, then interference to such normal movements and postures of the base of the neck, either through persistent lameness in the limbs or through constraints applied during training, may interfere with the maintenance of joint support in this region and lead to ongoing problems. Such loss of functionality seems especially likely if any of the normal supporting structures in the neck and cranial thorax are anomalous or lacking. The complex anatomy of the region suggests that there are many ways in which sufficient loading can occur to maintain function of all the tissues, especially the bones and joints. The complex interaction of these structures seems likely to provide predictable and sufficient support of the spinal column as long as free movement of the region is not interfered with and there is no source of pain causing the horse to interfere with the normal movements in the region.

## 5. Conclusions

This study demonstrated a uni-directional mechanical relationship between the equine forelimb, head and neck, which may have implications for balance and forelimb loading in the horse. These observations suggest that forelimb, neck and head movement coordination (observed in the horse during unrestrained locomotion) is significantly influenced by the mechanical linkages between body segments, rather than being entirely dependent on neurological input as previously thought. This highlights the potential significance of research directed at investigating passively induced movements in understanding common locomotory patterns. Additionally, it suggests a mode of postural control which may provide instantaneous adjustments to postural disturbances, thus promoting rapid and efficient locomotion.

## Figures and Tables

**Figure 1 animals-13-01894-f001:**
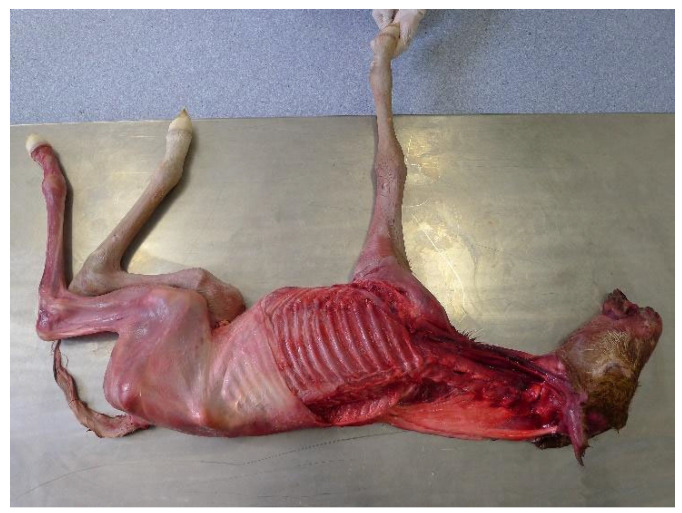
Foal placed in right lateral recumbency with the left forelimb and cervical musculature removed.

**Figure 2 animals-13-01894-f002:**
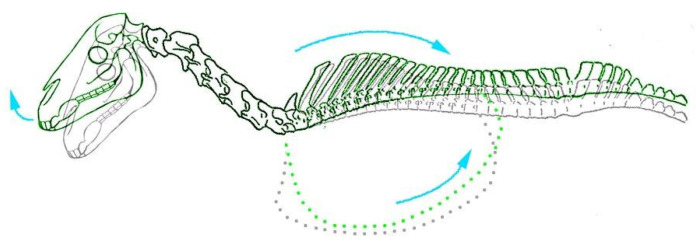
Caudodorsal movement of the thorax and extension of the head with forelimb retraction observed in equine foetuses (highlighted in green. Grey outline illustrates skeletal positioning in natural unrestrained position).

## Data Availability

Videos of related movements are available in the Appendix A.

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
