# Peer review of "Passive Dynamics of the Head, Neck and Forelimb in Equine Foetuses—An Observational Study"

_animals, 2023, doi:10.3390/ani13121894_

Round 1

Reviewer 1 Report

Dear Authors,

this is a well-designed study that provide an advancement of the current knowledge regarding our understanding of common locomotory patterns in the horse.

Thank you.

One comment on the experimental design:

The induced head movements were small in relation to large, forced leg movements. Thus, subjective manipulating of foal cadaver by the investigator to induce hypothesized head response may have affected outcome.

In live horse, legs are strictly bordered by the ground. In this study legs were freely moved, and thus the induced head movements may have been influenced by slightly changing the angle of the leg or by inducing more or less force on the leg by the investigator.

This limitation of the study could have been addressed a bid more clearly in the discussion.

Author Response

Reviewer 1

Thankyou for your kind comments. We have provided the responses to your comments

One comment on the experimental design: The induced head movements were small in relation to large, forced leg movements. Thus, subjective manipulating of foal cadaver by the investigator to induce hypothesized head response may have affected outcome.

Answer: As the videos demonstrate, the relationship between head, neck and forelimb movement was mostly evident with protraction and retraction of the entire limb rather than with small movements in the distal limb. As discussed in the paper, this is most likely due to the connection of the forelimb to the trunk via the serratus ventralis muscle. Although only a small head/ neck movement relative to forelimb, the relationship observed demonstrates the importance of this muscular connection which has been discussed in another equid study (Timing of head movements is consistent with energy minimization in walking ungulates David M. Loscher, Fiete Meyer, Kerstin Kracht and John A. Nyakatura Published:30 November 2016https://doi.org/10.1098/rspb.2016.1908). Also, the range of motion manipulated in the forelimbs of the cadavers, although large, is no larger than the range of motion we would see in a live horse at various gaits. Therefore, weI do not believe our movements are “inducing” an unrealistic relationship of movement

In live horse, legs are strictly bordered by the ground. In this study legs were freely moved, and thus the induced head movements may have been influenced by slightly changing the angle of the leg or by inducing more or less force on the leg by the investigator. This limitation of the study could have been addressed a bid more clearly in the discussion.

Answer: Yes these are good points and the following has been included at the end of in the discussion:

Admittedly, the effects of gravity and load bearing on the forelimb and their effects on the observed connectivity between the forelimb, head and neck were not able to be measured with this study design. However, given that there have already been studies investigating some degree of relationship between forelimb, head and neck movements in livinge, weight- bearing adult horses (47, 48, 49), we propose that the forward and backward motion of the head will persist when these forces are in play. This is something that could potentially be measured with a rostral head marker and motion cameras in future.

Reviewer 2 Report

This descriptive study focusses on the subjective observation of passive dynamics of equine foetuses post mortemin lateral recumbency, demonstrating the movement of head and neck when the forelimbs are moved forwards and backwards.  A uni-directional mechanical relationship between the forelimbs and the head and neck was found.

Unfortunately, the manuscript has a number of limitations. 

Manuscript writing: The hypothesis is not clear and changing throughout the manuscript àIntroduction “a uni-directional mechanical relationship exists between the head and limb”; Discussion “coordinated movement observed between the head, neck and forelimb in the horse during locomotion is largely dependent on the mechanical connectivity between segments rather than being entirely dependent on neurological input”. In my opinion, the latter can’t be established with the results of this study (please see below). 

Furthermore, in the discussion there is the mention of a first and a latter part of the hypothesis, which would need clarification in the manuscript. 

References: 

There are hardly any recent references (only one within the last 5 years). Despite this, in the text there is mention of recent work. 

For example, in the Introduction a third mode of postural stability focussing on passive dynamics of linked segments is described with references given from the years 1996, 2004, 1995 and 1967. 

Study design: 

Unfortunately, there is no quantitative analysis of the movements described. In the supplementary material there seems to be rulers positioned around the specimens but there is no mention of this in the methods or results. A subjective description only lacks scientific soundness and repeatability. 

The cadavers could have been their own control by comparing and measuring movements in left and right recumbency. There is brief mention of this in the discussion but unfortunately it is not presented in the results or explained in the methods. 

A quantitative analysis of the movements achieved by passive dynamics would have given information for future research and data to refer to for scientific soundness. This could have been achieved with the specimens positioned on a measurement grid or similar, for example. In the results it is stated that the greater range of axial skeletal movement was related to the caudal cervical vertebrae and cranial thorax – how was this measured?

The lack of defining a specific position of the foetus to start with raises the question if the movements observed would be reproducible. The foetus is positioned with the legs in an approximate standing position (Where there flexural deformities? What was the exact angle of the joints? Would this affect the observations?) and head and neck in a neutral position (which position was that? Did the position differ between subjects?) seems not reproducible and vague. 

Information on age (estimated age, or days post coitum) and breed of each individual subject would have been interesting. There was one foal, which died 24 hours post partum; was this foal mature and developed normally? Maturity, developmental abnormalities, size, weight, breed, gender and age might be important factors, when assessing for passive movement. More detailed information should be provided.

The conclusions drawn are interesting but are beyond the scope of this study. The translation from movement observation of cadaver foetuses in lateral recumbency to the adult horse in vivo should be viewed with care. It is therefore impossible, in my opinion, to draw conclusions on the movement coordination and motor control of body segments, particularly to which degree these movements are passive or dependent on neurological control in the live, adult horse. 

Author Response

Thankyou for your helpful comments.

Manuscript writing: The hypothesis is not clear and changing throughout the manuscript àIntroduction “a uni-directional mechanical relationship exists between the head and limb”; Discussion “coordinated movement observed between the head, neck and forelimb in the horse during locomotion is largely dependent on the mechanical connectivity between segments rather than being entirely dependent on neurological input”. In my opinion, the latter can’t be established with the results of this study (please see below). 

Furthermore, in the discussion there is the mention of a first and a latter part of the hypothesis, which would need clarification in the manuscript. 

Answer: We agree that we have not proved that the coordinated movement is largely dependent on the mechanical connectivity so we have substituted the words “significantly influenced” for the words “Largely dependent” in the abstract and the conclusion. 

The hypotheses have been rewritten and clarified as follows:

Therefore, it is hypothesized here that a uni-directional mechanical relationship exists between the head and limb. It is further hypothesized that the directions of related movements are consistent.

References: 

There are hardly any recent references (only one within the last 5 years). Despite this, in the text there is mention of recent work. 

For example, in the Introduction a third mode of postural stability focussing on passive dynamics of linked segments is described with references given from the years 1996, 2004, 1995 and 1967. 

Answer: You are right that many of the studies we refer to are not recent, so we have changed the wording appropriately. However, as we are discussing anatomy, most of our references are expected to be somewhat older as we reference lots of textbooks and early dissection works.

We have now included some more references as requested. However, we feel that the most important reference is the earliest reference that is germane to the ideas being presented. There are numerous recent papers in robotics and neurology that could be cited, but they are all somewhat peripheral to our central theme. We have tried to find either text books or papers that present the earliest publication of the ideas we are discussing.

Study design: 

Unfortunately, there is no quantitative analysis of the movements described. In the supplementary material there seems to be rulers positioned around the specimens but there is no mention of this in the methods or results. A subjective description only lacks scientific soundness and repeatability. 

Answer: The  large range of gestational age and condition of the foetuses as well as movement and drying of the water film underneath the specimens meant that absolute measurements of the size of movements were very variable. Repeated movements always created the same directional effects. Independent observers observed the same directional relationships and proportionate effects.

This explanation has now been added to the discussion.

An objective study need not have a numerical base. Numerous papers in the literature have a biphasic response (ie yes or no and in our case including extension or flexion). Our study is objective in that the answer of related movement is yes or no and the direction of movement is flexion or extension. Statistics were not included because the yes/no and flexion/extension observations were 100% consistent in all specimens in both left and right lateral recumbency. Hence the observations reported in our study should be easily repeatable by anyone with access to intact equine cadavers

The cadavers could have been their own control by comparing and measuring movements in left and right recumbency. There is brief mention of this in the discussion but unfortunately it is not presented in the results or explained in the methods. 

Answer: This was included in the results and we have now included mention of this in the methods and in the discussion.

A quantitative analysis of the movements achieved by passive dynamics would have given information for future research and data to refer to for scientific soundness. This could have been achieved with the specimens positioned on a measurement grid or similar, for example.

Answer: We tried various methods of standardising our measurements including a grid but felt the absolute measurements did not add to the results as explained above. We also observed that placing anything between the cadaver and the wet slippery table increased friction and reduced relative movements.

In the results it is stated that the greater range of axial skeletal movement was related to the caudal cervical vertebrae and cranial thorax – how was this measured?

Answer: This was a subjective observation backed up by independent observers. This point has now been added.

The lack of defining a specific position of the foetus to start with raises the question if the movements observed would be reproducible. The foetus is positioned with the legs in an approximate standing position (Where there flexural deformities? What was the exact angle of the joints? Would this affect the observations?) and head and neck in a neutral position (which position was that? Did the position differ between subjects?) seems not reproducible and vague. 

Answer: The intention was to look for the presence or absence of movement and the direction of that movement. Given the enormous number of possible variables this was considered to be the most appropriate method for this initial study. 

As you point out, the exact angle of the joints and head/neck was not measured. However, we think requiring this misses the point of the study. We are looking at connectivity between body segments through a range of motion. So regardless of where the joints start, they are all moving through a range of angles relative to the trunk in an extended position. The same goes with the head and neck. We were seeing flexion and extension of the head through a range. So unless we started to compare elevated or lowered neck positions or significantly flexed and extended neck positions, or variations in the thoracic position, we think approximating a neutral head position was not unreasonable.

Information on age (estimated age, or days post coitum) and breed of each individual subject would have been interesting. There was one foal, which died 24 hours post partum; was this foal mature and developed normally? Maturity, developmental abnormalities, size, weight, breed, gender and age might be important factors, when assessing for passive movement. More detailed information should be provided

Answer: There were no flexural deformities or other visible abnormalities in the specimens and this point has been included in the methods. The range of ages was approximately 80 to 340 days as stated in the methods and they were a mixture of sexes and apparently normal for their stage of develoment.

The conclusions drawn are interesting but are beyond the scope of this study. The translation from movement observation of cadaver foetuses in lateral recumbency to the adult horse in vivo should be viewed with care. It is therefore impossible, in my opinion, to draw conclusions on the movement coordination and motor control of body segments, particularly to which degree these movements are passive or dependent on neurological control in the live, adult horse

Answer: Despite the visual presence of such related movements in adult horses the hypothesis is presented as unproven, although supported by the consistency of the observations in this study. The study does give valuable insight into our understanding of movement coordination which may form the basis of future studies.

Reviewer 3 Report

translating the the role of the musculoskeletal system in acute and/or abrupt changes in position in adult horses based on the observations on these fetuses would be enhanced by demonstration of similar findings in yearlings or older horses. 

further discussion about the role of the fascial and muscular connections to signs that may be seen in adult horses such what a happens when there is an anomaly of the first rib and the impact of the changes that may have on the attachments of the seratis ventralis muscle

further integration of the role of neural inputs to the mechanical impact of the musculoskeletal  systems on stability and motion in  exercising horses would help the reader better understand how abnormalities in either system may impact performance 

Author Response

translating the the role of the musculoskeletal system in acute and/or abrupt changes in position in adult horses based on the observations on these fetuses would be enhanced by demonstration of similar findings in yearlings or older horses. 

Answer: We agree. We have observed such related movements in adult cadavers on the post mortem room floor, but the relative size and mass as well as the difficulty of reducing rigor mortis in any such massive specimen that is not very fresh makes such observations difficult and even more difficult to record or measure. Mention of such observations have been included in the discussion.

further discussion about the role of the fascial and muscular connections to signs that may be seen in adult horses such what a happens when there is an anomaly of the first rib and the impact of the changes that may have on the attachments of the seratis ventralis muscle

Answer: These are interesting potential problems which warrant further investigation

further integration of the role of neural inputs to the mechanical impact of the musculoskeletal  systems on stability and motion in  exercising horses would help the reader better understand how abnormalities in either system may impact performance 

Answer: It would be interesting to speculate on this subject, but we feel this is a difficult thing to measure and outside the scope of this study.

Round 2

Reviewer 2 Report

Thank you to the authors for answering to the points raised and amending the manuscript accordingly.

Unfortunately, in my opinion, the manuscript writing and study design remain with significant floors. To publishe a paper in a peer-reviewed journal, the manuscript would have to be re-written to clarify the intention, the limitations and the main points of the study. It is a concern, that the authors state in their answer to previous comments the following: “….the hypothesis is presented as unproven,…”. In the re-written introduction two hypotheses are stated: 1) there is a uni-lateral mechanical relationship between head and limb (in equine foetuses in vitro) and 2) the movements are consistent. Why in your opinion are these hypotheses unproven? Is it not the conclusions drawn from the results of the study that are “unproven”?

In the discussion the first sentence is: "This study set out to investigate the hypothesis that the coordinated movement observed between the head, neck and forelimb in the horse during locomotion is largely dependent on the mechanical connectivity between segments rather than being entirely dependent on neurological input."

If this is your hypothesis, then the study is not sufficient to investigate this! 

To understand the pattern of movement coordination this study could offer interesting information, which could be further investigated in future studies. But unfortunately the manuscript lacks the format to clearly present this information to the reader. 

Author Response

Reviewer comment: Unfortunately, in my opinion, the manuscript writing and study design remain with significant floors. To publishe a paper in a peer-reviewed journal, the manuscript would have to be re-written to clarify the intention, the limitations and the main points of the study. It is a concern, that the authors state in their answer to previous comments the following: “….the hypothesis is presented as unproven,…”. In the re-written introduction two hypotheses are stated: 1) there is a uni-lateral mechanical relationship between head and limb (in equine foetuses in vitro) and 2) the movements are consistent. Why in your opinion are these hypotheses unproven? Is it not the conclusions drawn from the results of the study that are “unproven”? In the discussion the first sentence is: "This study set out to investigate the hypothesis that the coordinated movement observed between the head, neck and forelimb in the horse during locomotion is largely dependent on the mechanical connectivity between segments rather than being entirely dependent on neurological input."If this is your hypothesis, then the study is not sufficient to investigate this!

 Response: We investigated this overarching hypothesis (so far unproven) by setting up the study to test the validity of the more specific related hypotheses 1. That there was a unidirectional movement between the limbs and the head and neck, and 2. That the relationship between the movements was consistent in all specimens. We have rewritten the first paragraph of the discussion to reflect this as follows (changes in italics):

This study set out to test the hypotheses that a uni-directional mechanical relationship exists between the equine head and forelimb and that this relationship is consistent. An inference from this would be that the coordinated movement observed between the head, neck and forelimb in the horse during locomotion is likely to be somewhat dependent on the mechanical connectivity between segments rather than being entirely dependent on neurological input. Results from this study confirmed the basic hypotheses by identifying synchronous uni-directional consistent locomotory patterns between the forelimb, head and neck in equine foetal foal cadavers. To our knowledge, this is the first example of unidirectional anatomically related movements which cannot be directly explained by reflexive neural feedback mechanisms. Instead, it is concluded here that coordinated forelimb, head and neck movement in the equine foetus can be achieved through passive dynamics.

Reviewer comment: To understand the pattern of movement coordination this study could offer interesting information, which could be further investigated in future studies. But unfortunately the manuscript lacks the format to clearly present this information to the reader. 

Response: We hope this is more clearly written now